# Learning Embeddings for Sequential Tasks Using Population of Agents

## Abstract

We present an information-theoretic framework to learn fixed-dimensional embeddings for tasks in reinforcement learning. We leverage the idea that two tasks are similar if observing an agent's performance on one task reduces our uncertainty about its performance on the other. This intuition is captured by our information-theoretic criterion which uses a diverse agent population as an approximation for the space of agents to measure similarity between tasks in sequential decision-making settings. In addition to qualitative assessment, we empirically demonstrate the effectiveness of our techniques based on task embeddings by quantitative comparisons against strong baselines on two application scenarios: predicting an agent's performance on a new task by observing its performance on a small quiz of tasks, and selecting tasks with desired characteristics from a given set of options.

## 1 Introduction

Embeddings are widely used to represent data points as vectors in a space that captures meaningful relations between them (Sun et al., 2014; Sung et al., 2018; Athar et al., 2020; Mikolov et al., 2013; Pennington et al., 2014; Cer et al., 2018; Zhang et al., 2021). They could also be utilized as representations for tasks, as studied in various areas such as multi-task learning (Zhang et al., 2018), meta-learning (Achille et al., 2019), and domain-adaptation (Peng et al., 2020).

In reinforcement learning (RL), task embeddings could be used to understand the shared structure in sequential decision-making problems if similar tasks are embedded in close proximity. Such embeddings could enable efficient, one-shot computation of task similarity, eliminating the need for time-consuming policy rollouts. Essentially, there is an underlying notion of skills required to solve sequential tasks, and several of these tasks require some skills in common. For instance, consider the tasks shown in Fig. 1. Each requires the agent to pick-up certain keys to unlock the door. The door in task $s_1$ requires the green key and the blue key, while the door in task $s_2$ requires the yellow key and the blue key. Thus, these tasks require the common skills of navigation, and picking the blue key.

Despite the potential benefits, prior work on learning task embeddings in RL (Qin et al., 2022; Schäfer et al., 2022; Arnekvist et al., 2018; Yoo et al., 2022; Rakelly et al., 2019; Bing et al., 2023; Gupta et al., 2018; Fu et al., 2020; Li et al., 2021; Lan et al., 2019; Walke et al., 2022; Sodhani et al., 2021b; Vuorio et al., 2019) does not explicitly optimize for task similarity. This could primarily be attributed to the lack of a general framework to measure (and reason about) similarities among sequential tasks.

To this end, we introduce an information-theoretic framework to learn fixed-dimensional embeddings for tasks in RL; the inner product in the embedding space captures similarity between tasks, and the norm of the embedding induces an ordering on the tasks based on their difficulties (see Fig. 1). A critical component of the framework is a population of agents exhibiting a diverse set of behaviors, which serves as an approximation for the space of agents. Our framework leverages the idea that two sequential tasks are similar to each other if observing the performance of an agent from this population on one task significantly decreases our uncertainty about its performance on the other. Concretely, we introduce an information-theoretic criterion to measure task similarity (Section 4.1), and an algorithm to empirically estimate it (Section 4.2). Through this, we construct a set of ordinal constraints on the embeddings (with each such constraint asserting the relative similarity between a triplet of tasks), and propose a training scheme for an embedding network to learn them (Section 4.3).

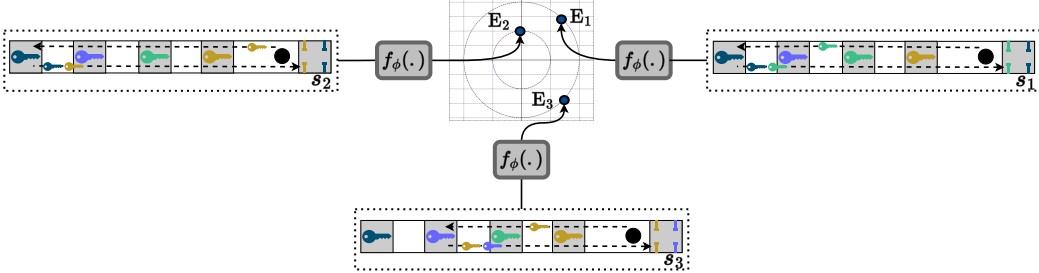

Figure 1: Schematics of our approach. We learn a task embedding function $f_\phi(.)$ that maps a task $s$ to its fixed-dimensional representation $\mathrm{E}$. In this illustration, we show the properties of the learned embeddings using the MULTIKEYNAV environment in which tasks require the agent (shown as a black circle) to pick-up certain keys (from the gray segments) to unlock the door (the right-most segment) that has certain requirements (shown in color in the form of gates). A possible solution trajectory is depicted using dotted lines. Keys on this trajectory correspond to the ones that the agent possesses at that point in time. For instance, in task $s_2$, the agent starts off with the yellow key in possession already. $\langle \mathrm{E}_1, \mathrm{E}_2 \rangle$ is greater than $\langle \mathrm{E}_1, \mathrm{E}_3 \rangle$, since tasks $s_1$ and $s_2$ have a common requirement of picking the blue key, and thus, are similar. Additionally, $\|\mathrm{E}_2\|_2$ is less than both $\|\mathrm{E}_1\|_2$ and $\|\mathrm{E}_3\|_2$, since task $s_2$ requires picking a single key, while tasks $s_1$ and $s_3$ require picking two keys, which makes them harder than $s_2$.

Besides assessing the learned embedding spaces through visualizations (Section 5), we ground our framework in two downstream scenarios that are inspired by real-world applications (Section 6). Firstly, we show the utility of our framework in predicting an agent's performance on a new task given its performance on a small quiz of tasks, which is similar to assessing a student's proficiency in adaptive learning platforms via a compact quiz (He-Yueya & Singla, 2021). Secondly, we demonstrate the application of our framework in selecting tasks with desired characteristics from a given set of options, such as choosing tasks that are slightly harder than a reference task. This is analogous to selecting desired questions from a pool for a personalized learning experience in online education systems (Ghosh et al., 2022). Through comparisons with strong baselines on a diverse set of environments, we show the efficacy of our techniques based on task embeddings.

To summarize, our work makes the following key contributions:

I. We introduce an information-theoretic framework to learn task embeddings in RL. As part of the framework, we propose a task similarity criterion which uses a diverse population of agents to measure similarity among sequential tasks (Sections 4.1 and 4.2).

II. We propose a scheme to learn task embeddings by leveraging the ordinal constraints imposed by our similarity criterion (Section 4.3).

III. To assess our framework, we perform visual assessments of the learned embedding spaces, and introduce two quantitative benchmarks: (a) agent's performance prediction, and (b) task selection with desired characteristics (Sections 5 and 6).

## 2 RELATED WORK

**Task embeddings in RL.** Several works in the meta-learning and multi-task learning literature have explored the use of task embeddings to model relationships between sequential tasks, where embeddings are either learned explicitly through objectives such as reconstruction (Arnekvist et al., 2018; Yoo et al., 2022; Bing et al., 2023) and trajectory-based contrastive learning (Fu et al., 2020; Li et al., 2021), or implicitly to aid generalization to new tasks (Lan et al., 2019; Walke et al., 2022; Sodhani et al., 2021b; Vuorio et al., 2019). While these methods integrate task embeddings with policies solely to improve performance, we propose a framework to learn general-purpose embeddings that can be used to quantify and analyze task similarities. Furthermore, in our framework, embedding computation is a one-shot operation, unlike prior work that relies on experience data from the policy for the task. These distinctions position our work as complementary to existing methods.

**Population-based techniques.** Our framework requires a diverse agent population. This is inline with (Furuta et al., 2021; Tylkin et al., 2021; Vinyals & et al., 2019; Jaderberg & et al., 2019; Parker-Holder et al., 2020), which use agent populations in the RL setting. For instance, Furuta et al. (2021) use a randomly generated agent population to empirically estimate *policy information capacity*, an information-theoretic measure of task difficulty in RL.

## 3 PROBLEM SETUP

**MDP and Tasks.** We use the Markov Decision Process (MDP) framework to define an environment. An MDP $\mathcal{M}$ is defined as a 6-tuple $(\mathcal{S}, \mathcal{A}, \mathcal{R}, \mathcal{T}, \mathcal{S}_{\text{init}}, \gamma)$, where $\mathcal{S}$ is the state space, $\mathcal{A}$ is the action space, $\mathcal{R} : \mathcal{S} \times \mathcal{A} \to \mathbb{R}$ is the reward function, $\mathcal{T} : \mathcal{S} \times \mathcal{S} \times \mathcal{A} \to [0, 1]$ is the transition dynamics, and $\mathcal{S}_{\text{init}} \subseteq \mathcal{S}$ is the set of initial states. Each state $s \in \mathcal{S}_{\text{init}}$ corresponds to a goal-based task (for example, the goal could be to reach a specific destination in a navigation task) where the reward is 0 on all transitions but those on which a task gets completed. On task completion, the reward is 1. As an alternative to discounting, at each time step, there is a failure probability of $1 - \gamma$, which incentivises the agent to solve the task quickly. This ensures that the cumulative reward is binary.

**Population of agents and task solvability.** $p(\theta)$ represents a distribution over the population of agents. Concretely, it is a distribution over the agents' policy parameters. We use $\Theta$ to represent the random variable that takes on the value $\theta$. $\mathcal{O}_{s,\Theta} \in \{0, 1\}$ is a Bernoulli random variable that takes on the value 1 if, on a rollout, an agent sampled from $p(\theta)$ could successfully solve the task $s \in \mathcal{S}_{\text{init}}$ (i.e., the cumulative reward is 1), and 0 otherwise. We call $\mathcal{O}_{s,\Theta}$ the optimality variable for task $s$. $\text{POS}(s) := \mathbb{E}[\mathcal{O}_{s,\Theta}]$ denotes the probability of success on $s$, and is the complement of task difficulty.

**Task embedding space.** Formally, we wish to learn a task embedding function (parameterized by $\phi$) $f_\phi : \mathcal{S}_{\text{init}} \to \mathbb{R}^n$, for an MDP $\mathcal{M}$ and a prior over the population of agents $p(\theta)$, that maps tasks to $n$-dimensional representations. The range of $f_\phi(.)$ is the task embedding space.

**Objective.** Our objective is to learn embeddings for sequential tasks with the following properties: (a) the inner product in the embedding space captures task similarity, where the realizations of optimality variables are similar for tasks that are embedded in close proximity, and (b) the norm of the embedding induces an ordering on the tasks based on their difficulties. We formalize these objectives in Section 4.

## 4 LEARNING FRAMEWORK

In Sections 4.1 and 4.2, we formally define our information-theoretic criterion to measure task similarity in RL and describe an algorithm to empirically estimate it. In Section 4.3, we view the problem of learning task embeddings through the lens of ordinal constraint satisfaction.

### 4.1 INFORMATION-THEORETIC MEASURE OF TASK SIMILARITY

Our goal is to measure similarity between sequential tasks. To this end, we propose the mutual information between task optimality variables as a measure of task similarity. This metric captures the intuition that two tasks are similar to each other if observing an agent's performance on one task reduces our uncertainty about its performance on the other. We begin by formally defining performance uncertainty. Thereafter, we provide a formal definition of our task similarity criterion.

**Definition 1** (Performance Uncertainty). *The entropy of the population with prior $p(\theta)$ solving a task $s$ is defined as:*

$$\mathcal{H}(\mathcal{O}_{s,\Theta}) = - \sum_{o \in \{0,1\}} \mathrm{P}(\mathcal{O}_{s,\Theta} = o) \log \mathrm{P}(\mathcal{O}_{s,\Theta} = o),$$

*where $\mathcal{O}_{s,\Theta}$ is the optimality variable for $s$.*

Thus, we could measure the similarity between two tasks $s_i, s_j \in \mathcal{S}_{\text{init}}$ as the reduction in $\mathcal{H}(\mathcal{O}_{s_i,\Theta})$ by observing $\mathcal{O}_{s_j,\Theta}$.

**Definition 2** (Task Similarity). *Given a prior over the population of agents $p(\theta)$, we measure the similarity between two tasks $s_i, s_j \in \mathcal{S}_{\text{init}}$ as the mutual information $\mathcal{I}(.;.)$ between their optimality*

variables $\mathcal{O}_{s_i,\Theta}$, $\mathcal{O}_{s_j,\Theta}$:

$$\mathcal{I}(\mathcal{O}_{s_i,\Theta}; \mathcal{O}_{s_j,\Theta}) = \mathcal{H}(\mathcal{O}_{s_i,\Theta}) - \mathcal{H}(\mathcal{O}_{s_i,\Theta} \mid \mathcal{O}_{s_j,\Theta}).$$

It quantifies the information obtained about $\mathcal{O}_{s_i,\Theta}$ by observing $\mathcal{O}_{s_j,\Theta}$.

## 4.2 EMPIRICAL ESTIMATION OF $\mathcal{I}$

We now outline an algorithm to empirically estimate $\mathcal{I}$. A comprehensive pseudocode detailing the computation of the criterion is provided in Appendix B. Given an MDP $\mathcal{M}$ and a prior distribution of the agent parameters $p(\theta)$, our algorithm uses $\mathtt{N}$ samples to estimate $\mathcal{I}(\mathcal{O}_{s_i,\Theta}; \mathcal{O}_{s_j,\Theta})$. For each sample, the algorithm randomly samples $\theta_l \sim p(\theta)$, and performs rollouts of $\pi_{\theta_l}$ from $s_i$ and $s_j$ to obtain estimates of the probability mass functions required for the computation of $\mathcal{I}$. The estimation procedure can be invoked with the signature ESTIMATE($s_i, s_j, \mathcal{M}, \pi, p(\theta), \mathtt{N}$).

## 4.3 LEARNING TASK EMBEDDINGS

---

**Algorithm 1** Learn the Task Embedding Function ($f_\phi$)

1: **procedure** TRAIN(Set of tasks $\mathcal{S}_{\text{init}}$, MDP $\mathcal{M}$, Policy $\pi$, Prior distribution of the agent parameters $p(\theta)$, Number of samples $\mathtt{N}$, Hyperparameter $\lambda$, Number of iterations $\mathtt{M}$)
2:     Initialize $\phi$.
3:     **for** $i \in \{1, \ldots, \mathtt{M}\}$ **do**
4:         Sample task $s_1, s_2, s_3 \sim \mathcal{S}_{\text{init}}$.
5:         $\mathrm{E}_1, \mathrm{E}_2, \mathrm{E}_3 \leftarrow f_\phi(s_1), f_\phi(s_2), f_\phi(s_3)$
6:         $\hat{\mathcal{I}}_{12} \leftarrow$ ESTIMATE($s_1, s_2, \mathcal{M}, \pi, p(\theta), \mathtt{N}$)
7:         $\hat{\mathcal{I}}_{13} \leftarrow$ ESTIMATE($s_1, s_3, \mathcal{M}, \pi, p(\theta), \mathtt{N}$)
8:         **if** $\hat{\mathcal{I}}_{12} > \hat{\mathcal{I}}_{13}$ **then**
9:             loss $\leftarrow \log(1 + \exp(\langle \mathrm{E}_1, \mathrm{E}_3 \rangle - \langle \mathrm{E}_1, \mathrm{E}_2 \rangle))$
10:        **else**
11:            loss $\leftarrow \log(1 + \exp(\langle \mathrm{E}_1, \mathrm{E}_2 \rangle - \langle \mathrm{E}_1, \mathrm{E}_3 \rangle))$
12:         Sample task $s_4, s_5 \sim \mathcal{S}_{\text{init}}$.
13:         $\mathrm{E}_4, \mathrm{E}_5 \leftarrow f_\phi(s_4), f_\phi(s_5)$
14:         **if** $\text{POS}(s_4) > \text{POS}(s_5)$ **then**
15:            loss $\leftarrow$ loss $+ \lambda \log(1 + \exp(\|\mathrm{E}_4\|_2 - \|\mathrm{E}_5\|_2))$
16:        **else**
17:            loss $\leftarrow$ loss $+ \lambda \log(1 + \exp(\|\mathrm{E}_5\|_2 - \|\mathrm{E}_4\|_2))$
18:         Update $\phi$ to minimize loss.
19:     **return** $\phi$

---

With the criterion to measure task similarity defined, we are interested in learning a task embedding function $f_\phi : \mathcal{S}_{\text{init}} \rightarrow \mathbb{R}^n$ (consequently, an embedding space) that satisfies the desiderata introduced in Section 3. To this end, we pose the problem of learning $f_\phi(.)$ as an ordinal constraint satisfaction problem. Essentially, the task similarity criterion $\mathcal{I}$ imposes a set $\mathcal{C}_{\text{MI}}$ of triplet ordinal constraints on the task embeddings. POS(.) imposes another set $\mathcal{C}_{\text{NORM}}$ of pairwise ordinal constraints.

Concretely, $\mathcal{C}_{\text{MI}}$ is a collection of ordered triplets of tasks s.t. for each $(s_1, s_2, s_3) \in \mathcal{C}_{\text{MI}}$, $\mathcal{I}(\mathcal{O}_{s_1,\Theta}; \mathcal{O}_{s_2,\Theta}) > \mathcal{I}(\mathcal{O}_{s_1,\Theta}; \mathcal{O}_{s_3,\Theta})$. Consequently, we would like to satisfy the constraint $\langle f_\phi(s_1), f_\phi(s_2) \rangle > \langle f_\phi(s_1), f_\phi(s_3) \rangle$. Likewise, $\mathcal{C}_{\text{NORM}}$ is a collection of ordered tuples of tasks s.t. for each $(s_1, s_2) \in \mathcal{C}_{\text{NORM}}$, $\text{POS}(s_1) > \text{POS}(s_2)$. Consequently, we would like to satisfy the constraint $\|f_\phi(s_2)\|_2 > \|f_\phi(s_1)\|_2$ (embeddings for easier tasks have smaller norm).

We learn the task embedding function $f_\phi(.)$, for an MDP $\mathcal{M}$ and a prior over the agent population $p(\theta)$, by optimizing the parameters $\phi$ to maximize the log-likelihood of the ordinal constraints under the Bradley-Terry-Luce (BTL) model (Luce, 1959). Concretely, given a triplet of tasks $(s_1, s_2, s_3)$, we define:

$$\mathrm{P}\big((s_1, s_2, s_3) \in \mathcal{C}_{\text{MI}}\big) := \frac{\exp\big(\langle f_\phi(s_1), f_\phi(s_2) \rangle\big)}{\exp\big(\langle f_\phi(s_1), f_\phi(s_2) \rangle\big) + \exp\big(\langle f_\phi(s_1), f_\phi(s_3) \rangle\big)}.$$

Similarly, given a tuple of tasks $(s_1, s_2)$, we define:

$$\mathrm{P}\big((s_1, s_2) \in \mathcal{C}_{\text{NORM}}\big) := \frac{\exp\big(\|f_\phi(s_2)\|_2\big)}{\exp\big(\|f_\phi(s_1)\|_2\big) + \exp\big(\|f_\phi(s_2)\|_2\big)}.$$

Hence, the task embedding function $f_\phi(.)$ is learned by solving the following optimization problem:

$$\min_\phi \left[ \mathop{\mathbb{E}}_{(s_1,s_2,s_3) \sim \mathcal{C}_{\text{MI}}} \log\Big(1 + \exp(\langle \mathrm{E}_1, \mathrm{E}_3 \rangle - \langle \mathrm{E}_1, \mathrm{E}_2 \rangle)\Big) + \lambda \mathop{\mathbb{E}}_{(s_4,s_5) \sim \mathcal{C}_{\text{NORM}}} \log\Big(1 + \exp(\|\mathrm{E}_4\|_2 - \|\mathrm{E}_5\|_2)\Big) \right],$$

where $\mathrm{E}_i$ denotes $f_\phi(s_i)$, and $\lambda$ is a hyperparameter. The pseudocode for the proposed algorithm to learn the task embedding function $f_\phi(.)$ is given in Algorithm 1.

| Environment | Task Variability | Action | State | Number of Tasks |
|---|---|---|---|---|
| MULTIKEYNAV | Reward Function | 7 | $\mathbb{R} \times \{0,1\}^6$ | Infinite |
| CARTPOLEVAR | Dynamics | 2 | $\mathbb{R}^5 \times \{0,1\} \times [200]$ | Infinite |
| POINTMASS | Dynamics | $\mathbb{R}^2$ | $\mathbb{R}^7$ | Infinite |
| KAREL | Reward Function + Dynamics | 52 | $\{0,1\}^{51840}$ | 73688 |
| BASICKAREL | Reward Function + Dynamics | 6 | $\{0,1\}^{88}$ | 24000 |

(a) Comparison of environments' complexity

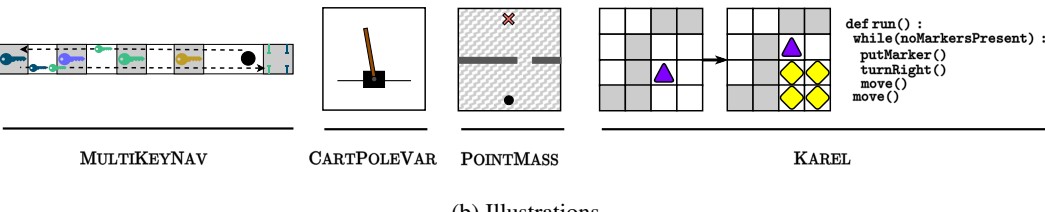

| MULTIKEYNAV | CARTPOLEVAR | POINTMASS | KAREL |

(b) Illustrations

Figure 2: We evaluate our framework on a diverse set of environments. (a) compares the characteristics of these environments. (b) illustrates these environments for a better understanding of the tasks.

# 5 EXPERIMENTS: VISUALIZATION OF EMBEDDING SPACES

In this section, we visualize the embedding spaces to gather qualitative insights, addressing the following research questions: (i) Can distinct clusters of tasks be identified by visualizing the embedding space? (ii) How does regularization through $\mathcal{C}_{\mathrm{NORM}}$ affect the embedding space? (iii) What influence do agent population and environment specification have on the embedding space? We begin by discussing the rationale for environment selection, describing these environments. Subsequently, we provide an overview of the embedding networks' training process, followed by the qualitative results.

## 5.1 ENVIRONMENTS

We evaluate our framework on environments with diverse characteristics to demonstrate its generality and scalability to different sequential decision-making problems (see Fig. 2). As the running example, we use MULTIKEYNAV (based on (Devidze et al., 2021)) because of its compositional nature in which the agent needs to compose different actions for picking keys (with four distinct key types, each requiring a specific action to be picked) in a task-specific manner to unlock the door. This also makes it suitable for ablation experiments. Task variability comes from the agent's initial position, the keys that it possesses initially, and the door type (with each type requiring a unique set of keys).

Given that task variability in MULTIKEYNAV comes from the reward function, we use CARTPOLE-VAR to highlight our framework's applicability to environments where it comes from the dynamics instead. This environment is a variation of the classic control task from OpenAI gym (Brockman et al., 2016), and also takes inspiration from (Sodhani et al., 2021a) in which the forces applied by each action could be negative as well. Tasks in this environment require keeping a pole attached by an unactuated joint to a cart upright for 200 timesteps by applying forces to the left (action 0) or to the right (action 1) of the cart. Task variability comes from the force F applied on the cart by each action, and the TaskType $\in \{0, 1\}$. Tasks of Type 0 involve "Pulling" with action 0 pulling the cart from the left and action 1 pulling the cart from the right, while tasks of Type 1 involve "Pushing".

We select POINTMASS (introduced in (Klink et al., 2020)) to test if our framework can handle continuous action spaces. In this environment, the agent applies forces to control a point mass inside a walled square. Tasks require reaching a fixed goal position through a gate, with task variability arising from the gate width and position, along with the coefficient of kinetic friction of the space.

Finally, to investigate our framework's scalability, we use the real-world environment KAREL from (Bunel et al., 2018), which is a challenging environment with applications in programming education. Tasks in this environment require the agent to synthesize a program, potentially containing control flow constructs such as loops and conditionals, satisfying a given specification comprising input-output examples. This program serves as a controller for an avatar navigating a grid, where each cell could

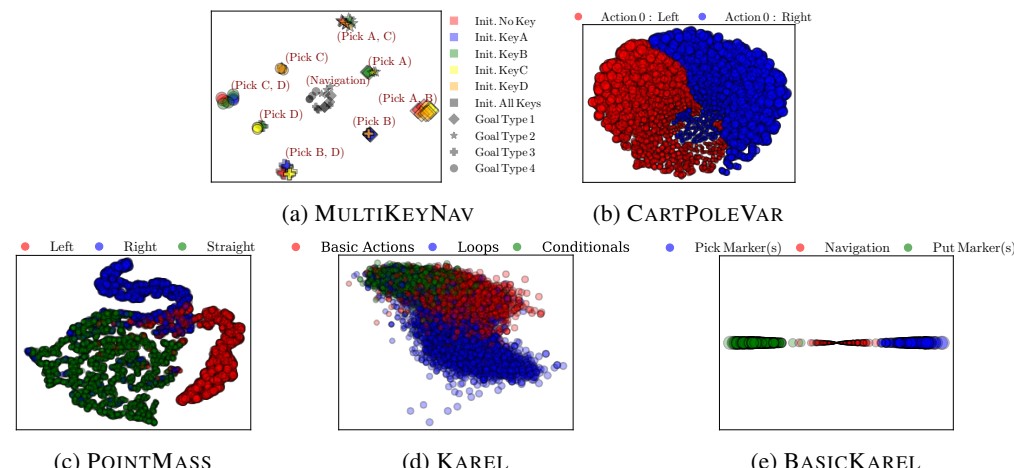

Figure 3: Visualization of the task embedding spaces learnt through our framework. Each point represents a task, and the size of the points is proportional to the norm of the embeddings.

contain marker(s), correspond to a wall, or be empty. The avatar can traverse the grid and manipulate it by picking or placing markers. Thus, an example in the specification comprises the `Pre-Grid` and the corresponding `Post-Grid`. In addition, we evaluate our framework on BASICKAREL (Tzannetos et al., 2023), which is a simpler variant of KAREL that excludes control flow constructs.

## 5.2 TRAINING PROCESS

To learn the task embedding function, we first obtain the agent population by taking snapshots while training a neural network policy using either behavioral cloning (Bain & Sammut, 1995) or policy gradient methods (Sutton et al., 1999). Concretely, a snapshot is recorded if the average performance on a validation set of tasks (denoted as $\mathcal{S}_{\mathrm{snap}}$) improves by $\delta_{\mathrm{snap}}$ compared to the previously recorded snapshot. A snapshot of the untrained policy is recorded by default. Different subpopulations, obtained by either masking actions or by using biased task distributions during training, are combined to form the final population. Here, masking a certain action corresponds to setting its logit to a large negative number. Using biased task distribution during training is another way to inject diversity into the population. In MULTIKEYNAV, for instance, using a biased task distribution could correspond to assigning low probability mass to tasks with certain types of doors in the initial state distribution during training. Finally, we parameterize the task embedding function $f_\phi(.)$ with a neural network, optimizing its parameters as described in Algorithm 1. We provide additional details in Appendix E.

## 5.3 VISUALIZATIONS AND QUALITATIVE RESULTS

We visualize the embedding spaces on a 2-dimensional map using t-SNE (van der Maaten & Hinton, 2008) to identify distinct clusters of tasks. Although t-SNE preserves the local structure, it does not necessarily preserve the embeddings' norm. For this reason, we scale the points in proportion to the norm of the embeddings. Additionally, we provide PCA plots in Appendix G.

**Visualizations.** For MULTIKEYNAV (Fig. 3a), our framework discovers distinct clusters of tasks, with each cluster corresponding to a unique set of keys that need to be picked. The norm of the embeddings is in accordance with the number of keys that need to be picked (with tasks requiring navigation only having the smallest norm). Additionally, tasks in clusters adjacent to each other share a common key requirement. For CARTPOLEVAR (Fig. 3b), our framework discovers that each task exhibits one of two types of underlying dynamics. In one (+ve F and Type 0, or −ve F and Type 1), action 0 moves the cart to the left, while in the other (−ve F and Type 0, or +ve F and Type 1), action 0 moves the cart to the right. For POINTMASS (Fig. 3c), our framework discovers three clusters of tasks based on the behavior that the agent needs to exhibit near the gate. The first cluster includes tasks in which the agent need not steer to cross the gate, while the second and third clusters contain tasks in which the agent must steer left or right to cross the gate, respectively. For KAREL and BASICKAREL (Fig. 3d and 3e), our framework discovers different clusters of tasks based on

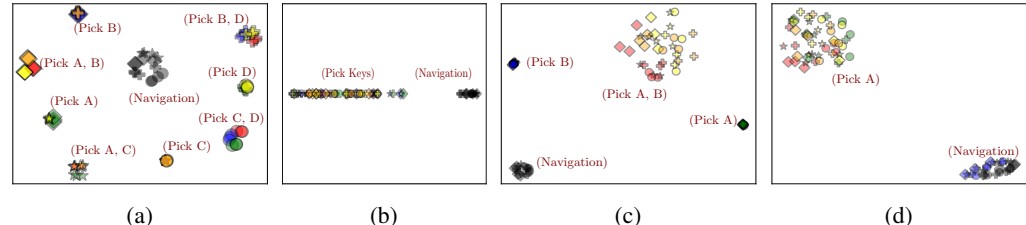

|       (a)        |       (b)        |       (c)        |       (d)        |

Figure 4: Task embedding spaces for the MULTIKEYNAV environment: (a) without $\mathcal{C}_{\mathrm{NORM}}$, (b) `pickKey` actions masked, (c) all doors require `KeyA`, `KeyB`, and (d) all doors require `KeyA`.

whether the solution code requires loops or conditionals, and whether the agent needs to pick or put markers in the grid, respectively.

**Ablation w.r.t. $\mathbf{C_{NORM}}$.** Fig. 4a shows the task embedding space learned without the norm ordinal constraints $\mathcal{C}_{\mathrm{NORM}}$ (i.e., $\lambda$ is set to 0). As expected, the norm of the embeddings is not proportional to the number of keys that need to be picked. Instead, the points are nearly uniform in size.

**Ablation w.r.t. population specification.** To understand the effect of population on the task embedding space, we learn the embedding function $f_\phi(.)$ for MULTIKEYNAV using an agent population in which `pickKey` actions are masked (Fig. 4b). In this case, we obtain two distinct clusters of tasks – one of the clusters contains tasks that cannot be solved (these tasks require picking key(s)), and the other contains tasks that require navigation only. These results emphasize the importance of the population's quality in learning a good task embedding space.

**Ablation w.r.t. environment specification.** In this ablation experiment, we change the environment specification and check its impact on the task embedding space. Concretely, we learn the embedding space for the following variants of MULTIKEYNAV: (a) each door requires `KeyA` and `KeyB` (Fig. 4c), i.e., all the doors have identical key requirements, and (b) each door requires `KeyA` only (Fig. 4d). Modifying the environment specification changes the task semantics, thereby impacting the task embedding space. Thus, these results are inline with our intuition.

## 5.4 COMPARISON WITH EXISTING WORK

To compare our framework with existing methods, we introduce *PredModel* baseline (inspired by prior work) and use silhouette scores based on the intuitively identified clusters of tasks to measure clustering quality in the learned embedding spaces. We also compare our method against embedding networks with random weights (*RandomModel*).

| Environment | RandomModel | PredModel | Ours |
|---|---|---|---|
| MULTIKEYNAV | $0.036 \pm 0.048$ | $-0.037 \pm 0.003$ | $\mathbf{0.753 \pm 0.001}$ |
| CARTPOLEVAR | $0.015 \pm 0.016$ | $0.242 \pm 0.007$ | $\mathbf{0.325 \pm 0.009}$ |
| POINTMASS | $0.104 \pm 0.026$ | $-0.010 \pm 0.004$ | $\mathbf{0.380 \pm 0.019}$ |
| BASICKAREL | $-0.058 \pm 0.007$ | $-0.002 \pm 0.003$ | $\mathbf{0.811 \pm 0.019}$ |

Figure 5: Comparison of silhouette scores (higher is better) based on intuitively identified clusters of tasks in the learned embedding spaces. The scores for our models are consistently better.

Most existing methods (e.g., PEARL (Rakelly et al., 2019)) utilize variational inference to learn latent context from task-specific experience data, where the inference network could be trained to reconstruct the MDP for the task through predictive models of reward and dynamics. To adapt this approach to our setting, we connect our formalism of tasks as initial states to the contextual MDP setting (Hallak et al., 2015), where each context (e.g., MULTIKEYNAV's context: agent's initial position, possessed keys initially, door type) corresponds to a distinct task represented by a separate MDP with context-dependent transitions and rewards. This set of MDPs can be converted into an equivalent MDP by including context variables as part of the state. In this converted MDP, each initial state represents a task, as it determines the context for the entire episode. The context is observable.

The modifications needed for the *PredModel* baseline are as follows: Firstly, since context is observable in our setup, we condition the approximate posterior over the embeddings on the initial state, eliminating the need for experience data. Secondly, we train the predictive models on states with context variables removed, ensuring the utilization of the task embedding that the model is conditioned on. We provide additional technical details in Appendix D.

**Results.** Fig. 5 reports the silhouette scores, averaged across 3 random seeds, with 1000 tasks per seed (5000 for BASICKAREL). The scores for the models learned through our framework are consistently better. While the *PredModel* baseline clusters similar tasks together in the embedding space for CARTPOLEVAR, it fails to do so in rest of the environments. In contrast to CARTPOLEVAR, where task variability comes from dense differences in the dynamics, task variability in other environments comes from sparse differences in the reward function and/or dynamics. Therefore, we hypothesize that the *PredModel* baseline fails on environments with sparse variability across tasks.

## 6 EXPERIMENTS: APPLICATION SCENARIOS

In this section, we evaluate our framework on two application scenarios: performance prediction, and task selection. We conduct this evaluation on MULTIKEYNAV and CARTPOLEVAR, as they cover two distinct sources of task variability, namely reward function and dynamics.

### 6.1 PERFORMANCE PREDICTION

First, we assess the learned task embeddings by using them to predict an agent's performance on a task $s_{\text{test}} \in \mathcal{S}_{\text{init}}$ after observing its performance on a quiz $\mathcal{S}_{\text{quiz}} \subseteq \mathcal{S}_{\text{init}}$. Specifically, we seek to answer the following research question: Would an agent show similar performance on tasks that are close to each other in the learned task embedding space? We begin by creating a benchmark for this application scenario, and then compare our technique against various baselines.

**Benchmark.** Formally, given the realizations of the task optimality variables of a set of tasks for an agent $\theta$, we are interested in predicting the most probable realization of the task optimality variable of a new task for the same agent without observing $\theta$. To create benchmarks for this scenario, we generate datasets for quiz sizes ranging from 1 to 20, with 5000 examples for both training and testing. Each example is generated by randomly sampling a quiz $\mathcal{S}_{\text{quiz}}$ of desired size, along with a task $s_{\text{test}}$ from $\mathcal{S}_{\text{init}}$, and then recording the performance of an agent $\theta$, sampled from the population, on these tasks. Performance prediction techniques are evaluated on this benchmark by measuring prediction ac-

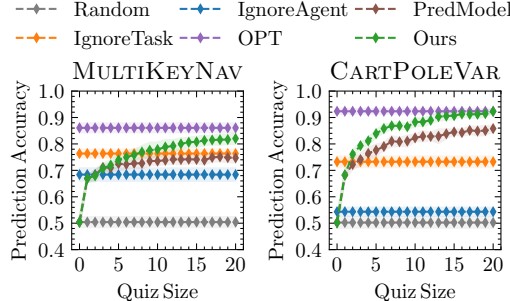

Figure 6: Results for performance prediction using task embeddings. Our technique (listed as *Ours*) is competitive with the *OPT* baseline, which is the best one could do on this benchmark.

curacy on the test examples. The techniques are evaluated on each dataset by partitioning it into 10 folds and reporting the mean prediction accuracy across the folds along with the standard error.

**Our approach.** Our prediction technique performs soft-nearest neighbor matching of $s_{\text{test}}$ with $\mathcal{S}_{\text{quiz}}$ in the task embedding space to predict performance on $s_{\text{test}}$. Concretely, given the embedding function $f_\phi(.)$, the prediction is $\mathbb{1}_{c>0.5}$, where c equals $\frac{\sum_{s \in \mathcal{S}_{\text{quiz}}} o_s \exp(-\beta \|f_\phi(s) - f_\phi(s_{\text{test}})\|_2^2)}{\sum_{s \in \mathcal{S}_{\text{quiz}}} \exp(-\beta \|f_\phi(s) - f_\phi(s_{\text{test}})\|_2^2)}$, $o_s$ is the realization of the task optimality variable for task $s$, and $\beta$ is a hyperparameter.

**Baselines.** Besides *PredModel*, we compare against different levels of oracle knowledge: (i) *Random*: Randomly predicts the agent's performance. (ii) *IgnoreTask*: Predicts the agent to succeed on $s_{\text{test}}$ iff the probability that it succeeds on a random task exceeds 0.5. (iii) *IgnoreAgent*: Predicts the agent to succeed on $s_{\text{test}}$ iff the probability that a random agent succeeds on it exceeds 0.5. (iv) *OPT*: Predicts the agent to succeed on $s_{\text{test}}$ iff the probability that it succeeds on $s_{\text{test}}$ exceeds 0.5.

**Results.** Fig. 6 shows the prediction accuracies of various techniques. Our method is competitive with the *OPT* baseline, which provides an upper-bound on the prediction accuracy but relies on the unrealistic assumption of full observability of both the agent and task.

### 6.2 TASK SELECTION

Next, we assess the learned embeddings by using them to select tasks with desired characteristics. Specifically, we seek to answer the following research questions: (i) Does the inner product in the learned task embedding space capture task similarity according to our information-theoretic criterion? (ii) Does the norm of the embedding learned by our framework induce an ordering on the tasks

Figure 7: Results for task selection using task embeddings (dark bars represent Top-3 accuracy and light bars represent Top-1). Our technique (listed as *Ours*) is competitive with $\widehat{OPT}_{50}$. Further, it outperforms $Ours_{woNorm}$ on Type-2 queries, highlighting the significance of $\mathcal{C}_{\mathrm{NORM}}$ in our framework.

based on their difficulties? We begin by creating a benchmark for this application scenario, and then compare our technique for task selection using task embeddings against various baselines.

**Benchmark.** Amongst several options of tasks $\mathcal{S}_{\mathrm{options}}$, we are interested in choosing the task that best matches the desired characteristics, which we categorize into two query types: *Type-1*: *Select the task that is the most similar to a given reference task* $s_{\mathrm{ref}}$. The ground-truth answer to this query is $\arg\max_{s \in \mathcal{S}_{\mathrm{options}}} \mathcal{I}(\mathcal{O}_{s_{\mathrm{ref}},\Theta}; \mathcal{O}_{s,\Theta})$. *Type-2*: *Select the task that is the most similar to (but harder than) a given reference task* $s_{\mathrm{ref}}$. Out of all the tasks in $\mathcal{S}_{\mathrm{options}}$ that are harder than $s_{\mathrm{ref}}$, the ground-truth answer to this query is the task most similar to it. To create benchmarks for this scenario, we generate a dataset of 50 examples. Each example consists of a randomly sampled $s_{\mathrm{ref}}$ and 10 tasks that form $\mathcal{S}_{\mathrm{options}}$. Additionally, each benchmark includes 5 easy tasks for reference (determined by ranking a randomly sampled pool of 500 tasks). We evaluate task selection techniques by reporting mean selection accuracy across 4 randomly sampled datasets, along with the standard error.

**Our approach.** We use task embeddings to rank the options according to similarity and/or difficulty, based on which the selection is made. We additionally compare our technique based on task embeddings learned without $\mathcal{C}_{\mathrm{NORM}}$ (listed as $Ours_{woNorm}$).

**Baselines.** Besides *PredModel*, we compare against the following baselines: (i) *Random*: Randomly selects answers from $\mathcal{S}_{\mathrm{options}}$. (ii) *StateSim*: Measures task similarity based on state representation distances. For queries of type 2, it considers a task $s_1$ to be harder than $s_2$ iff the similarity between $s_1$ and the task most similar to it in the set of easy reference tasks, is less than that for $s_2$. (iii) *TrajectorySim*: Measures task similarity using the edit distance between expert trajectories. (iv) $OPT$: Estimates task similarity and difficulty using the entire agent population. Given the variance in the estimation process, this is the best one could do on this benchmark. (v) $\widehat{OPT}_{50}$: Estimates task similarity and difficulty using a randomly sampled 50% of the population.

**Results.** Fig. 7 compares different techniques' selection accuracies on the task selection benchmark. Our technique outperforms *Random*, *StateSim*, *TrajectorySim*, and *PredModel*, and is competitive with $\widehat{OPT}_{50}$. This suggests that the inner product in the learned task embedding space successfully captures similarity between tasks. Notably, our technique significantly outperforms $Ours_{woNorm}$ on Type-2 queries, indicating that the norm of the embedding effectively orders tasks by difficulty.

## 7 CONCLUSION

In this work, we introduced an information-theoretic framework for learning task embeddings in sequential decision-making settings. Through experiments on diverse environments, we empirically demonstrated that the inner product in the embedding space captures task similarity, and the norm of the embedding induces an ordering on the tasks based on their difficulties. A limitation of our current framework is the requirement for tasks to be goal-based, which we plan to address in future work. This could involve using the difference between the cumulative reward obtained during the rollout and the maximum achievable cumulative reward for the given task to parameterize the Bernoulli optimality variable. Additionally, the agent population plays a crucial role in our framework, and it would be interesting to explore more principled methods for construction that explicitly optimize for diversity. Further, empirically estimating the proposed similarity criterion by directly estimating the underlying mass functions could be sample-inefficient for some environments. Therefore, a promising direction is to construct sample-efficient estimators for it. Moreover, evaluation in multi-agent settings, where the task embedding could encode the behavior of non-ego agents, is another interesting direction.

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

# A    TABLE OF CONTENTS

In this section, we briefly describe the content provided in the paper's appendices.

# B    PSEUDOCODE FOR EMPIRICAL ESTIMATION OF TASK SIMILARITY

The pseudocode for the proposed algorithm to empirically estimate $\mathcal{I}$ is given in Algorithm 2. Given an MDP $\mathcal{M}$ and a prior distribution of the agent parameters $p(\theta)$, the algorithm uses $\mathbb{N}$ samples to estimate $\mathcal{I}(\mathcal{O}_{s_i,\Theta}; \mathcal{O}_{s_j,\Theta})$. For each sample, the algorithm randomly samples $\theta_l \sim p(\theta)$, and performs rollouts of $\pi_{\theta_l}$ from $s_i$ and $s_j$ to obtain estimates of the probability mass functions required for the computation of $\mathcal{I}$. Note that $\mathcal{H}_b(p)$ computes the entropy of a Bernoulli random variable $X$ s.t. $X$ takes the value 1 with probability $p$.

---

**Algorithm 2** Empirically Estimate Task Similarity ($\mathcal{I}$)

---

1: **procedure** ESTIMATE(Task $s_i$, Task $s_j$, MDP $\mathcal{M}$, Policy $\pi$, Prior distribution of the agent parameters $p(\theta)$, Number of samples $\mathbb{N}$)
2:     $\texttt{n\_i} \leftarrow 0$                                                                  ▷ #successes on $s_i$
3:     $\texttt{n\_j} \leftarrow 0$                                                                  ▷ #successes on $s_j$
4:     $\texttt{n\_i\_j\_1} \leftarrow 0$                                                  ▷ #successes on $s_i$ given success on $s_j$
5:     $\texttt{n\_i\_j\_0} \leftarrow 0$                                                  ▷ #successes on $s_i$ given failure on $s_j$
6:     **for** $l \in \{1, \ldots, \mathbb{N}\}$ **do**
7:         Sample agent parameters $\theta_l \sim p(\theta)$ and set it to $\pi$.
8:         Perform a rollout of $\pi_{\theta_l}$ from $s_i$ on $\mathcal{M}$.
9:         Perform a rollout of $\pi_{\theta_l}$ from $s_j$ on $\mathcal{M}$.
10:         **if** rollout from $s_i$ is a success **then**
11:             $\texttt{n\_i} \leftarrow \texttt{n\_i} + 1$
12:             **if** rollout from $s_j$ is a success **then**
13:                 $\texttt{n\_i\_j\_1} \leftarrow \texttt{n\_i\_j\_1} + 1$
14:             **else**
15:                 $\texttt{n\_i\_j\_0} \leftarrow \texttt{n\_i\_j\_0} + 1$
16:         **if** rollout from $s_j$ is a success **then**
17:             $\texttt{n\_j} \leftarrow \texttt{n\_j} + 1$
18:     $\hat{\mathcal{I}} \leftarrow \mathcal{H}_b(\frac{\texttt{n\_i}}{\mathbb{N}}) - (\frac{\texttt{n\_j}}{\mathbb{N}})\mathcal{H}_b(\frac{\texttt{n\_i\_j\_1}}{\texttt{n\_j}}) - (1 - \frac{\texttt{n\_j}}{\mathbb{N}})\mathcal{H}_b(\frac{\texttt{n\_i\_j\_0}}{\mathbb{N}-\texttt{n\_j}})$
19:     **return** $\hat{\mathcal{I}}$

---

# C    ENVIRONMENT DETAILS

## C.1    MULTIKEYNAV

This environment corresponds to a navigation task in a one-dimensional line segment $[0, 1]$, where the agent has to pick certain keys using appropriate `pickKey` actions (one action for each key type) and unlock the door located towards the right. A task in this environment is considered to be solved if the agent successfully unlocks the door. The environment used in our experiments is based on the work of Devidze et al. (2021); however, we adapted it to have multiple keys that need to be picked.

More concretely, there are four keys, `KeyA`, `KeyB`, `KeyC`, and `KeyD`, located on the segments $[0, 0.1]$, $[0.2, 0.3]$, $[0.4, 0.5]$, $[0.6, 0.7]$, respectively. A door is located on the segment $[0.9, 1]$. The door could

be of the following 4 types: Type 1 (00), Type 2 (01), Type 3 (10), or Type 4 (11). Doors of Type 1 require `KeyA` and `KeyB`, doors of Type 2 require `KeyA` and `KeyC`, doors of Type 3 require `KeyB` and `KeyD`, and doors of Type 4 require `KeyC` and `KeyD`.

The set of initial states, $\mathcal{S}_{\text{init}}$, is the same as the set of states $\mathcal{S}$. Each state $s \in \mathcal{S}$ corresponds to a 7-tuple (`Location`, `KeyStatusA`, `KeyStatusB`, `KeyStatusC`, `KeyStatusD`, `doorBit1`, `doorBit2`). Here, `Location` denotes the agent's location on the line segment, and `KeyStatusA`, `KeyStatusB`, `KeyStatusC`, `KeyStatusD` are flags for whether the agent has picked up the corresponding key. Task variability in this environment comes from the agent's initial position, the keys that it possesses initially, and the door type (with each type requiring a unique set of keys).

The action space is $\mathcal{A} = \{$`moveLeft`, `moveRight`, `pickKeyA`, `pickKeyB`, `pickKeyC`, `pickKeyD`, `finish`$\}$. `moveLeft` and `moveRight` move the agent across the environment with step size $0.075 + \epsilon$, where $\epsilon \sim U(-0.01, 0.01)$. If `pickKeyA` is executed at a location that lies on the segment containing `KeyA`, `KeyStatusA` becomes `True`, else the environment crashes. Likewise for `pickKeyB`, `pickKeyC`, and `pickKeyD`. The agent gets a reward of 1 on executing `finish` if it is at a location that lies on the segment containing the door and possesses the required keys; `finish` results in a crash otherwise. The horizon length is 40 and $\gamma$ is 0.999.

## C.2 CARTPOLEVAR

This environment is a variation of the classic control task from OpenAI gym (Brockman et al., 2016), and also takes inspiration from (Sodhani et al., 2021a) in which the forces applied by each action could be negative as well. It consists of a pole (mass: 0.1 kg, length: 1 m) attached by an unactuated joint to a cart (mass: 1 kg), which moves along a track. The agent controls the cart in the presence of gravity (g: 9.8 m/s$^2$) by applying forces to the the left (action 0) or to the right (action 1) of the cart. A task is considered to be solved if the agent keeps the pole upright for 200 timesteps.

Each state $s \in \mathcal{S}$ corresponds to a tuple (x, v, $\theta$, $\omega$, F, `TaskType`, `NumSteps`). Here, x denotes the position of the cart, v denotes the velocity of the cart, $\theta$ denotes the angle that the pole makes with the vertical, $\omega$ denotes the angular velocity of the pole, $F \in [-15\,\text{N}, -5\,\text{N}] \cup [5\,\text{N}, 15\,\text{N}]$ denotes the force applied on the cart by each action, `TaskType` $\in \{0, 1\}$ denotes the type of the task, and `NumSteps` denotes the number of steps passed since the beginning of the episode.

Task variability in this environment comes from the force F applied on the cart by each action, and the `TaskType` $\in \{0, 1\}$. Tasks of Type 0 involve "Pulling" with action 0 pulling the cart from the left and action 1 pulling the cart from the right, while tasks of Type 1 involve "Pushing". At any timestep, if $\theta \notin [-12°, 12°]$, the pole is not upright and consequently, the environment crashes. The horizon length is 200 and $\gamma$ is 1.

## C.3 POINTMASS

This environment was introduced by Klink et al. (2020). We provide details here for completeness.

The agent applies forces to control a point mass inside a square space $[-4, 4] \times [-4, 4]$ surrounded by walls. The space exhibits friction, with the coefficient of kinetic friction $\mu_{\text{k}} \in [0, 4]$. Additionally, there is a gate of width $w_{\text{g}} \in [-4, 4]$ at position $p_{\text{g}} \in [0.5, 8]$, effectively spanning the segment $[p_{\text{g}} - 0.5 w_{\text{g}}, p_{\text{g}} + 0.5 w_{\text{g}}]$. The agent always starts off from the fixed initial position $[0, 3]$. A task in this environment is considered to be solved if the point mass reaches the fixed goal position $[0, -3]$, which requires crossing the gate.

Each state $s \in \mathcal{S}$ corresponds to a tuple (x, $v_{\text{x}}$, y, $v_{\text{y}}$). Here, $[x, y]$ denotes the position of the point mass, while $[v_{\text{x}}, v_{\text{y}}]$ denotes the velocity. The actions are $[F_x, F_y] \in [-10, 10] \times [-10, 10]$, where $F_x$ and $F_y$ correspond to forces applied along the $x$ and the $y$ axis, respectively. Task variability in this environment comes from the width $w_{\text{g}}$ and the position $p_{\text{g}}$ of the gate, along with the coefficient of kinetic friction $\mu_{\text{k}}$ of the space. At any timestep, if the point mass crashes into the wall, the environment crashes. The horizon length is 100 and $\gamma$ is 0.99.

## C.4 KAREL

This is the Karel program synthesis environment from (Bunel et al., 2018). Karel is an educational programming language widely used in introductory CS courses. The environment consists of an avatar (characterized by its position and orientation) inside an $18 \times 18$ grid in which each cell could contain up to 10 markers, correspond to a wall, or be empty. The avatar can move inside the grid and modify it by picking or placing markers. The objective of each task is to synthesize the program $\pi^*$ (which is a controller for the avatar) in the Karel domain-specific language (DSL) given 5 input-output examples for it in the form of `Pre-Grid` and its corresponding `Post-Grid`. A task is considered to be solved if the synthesized program $\pi$ generalizes to a held-out test example for $\pi^*$.

The Karel DSL is shown in Figure 8. Task variability in this environment comes from the set of input-output examples. The state in this environment comprises of the program specification (i.e., the input-output examples) and the partial program synthesized so far. The tokens of the DSL form the action space. The horizon length is 24 and $\gamma$ is 1. We use a set of 73688 tasks (with the number of tokens in $\pi^*$ ranging between 10 and 14) sampled from the dataset used in (Bunel et al., 2018) (accessible at https://msr-redmond.github.io/karel-dataset/).

$$
\begin{aligned}
\text{Prog } p \quad &:= \quad \texttt{def run() : } s \\
\text{Stmt } s \quad &:= \quad \texttt{while}(b) : s \mid \texttt{repeat}(r) : s \mid s_1; s_2 \mid a \\
&\quad \mid \quad \texttt{if}(b) : s \mid \texttt{ifelse}(b) : s_1 \texttt{ else} : s_2 \\
\text{Cond } b \quad &:= \quad \texttt{frontIsClear()} \mid \texttt{leftIsClear()} \mid \texttt{rightIsClear()} \\
&\quad \mid \quad \texttt{markersPresent()} \mid \texttt{noMarkersPresent()} \mid \texttt{not } b \\
\text{Action } a \quad &:= \quad \texttt{move()} \mid \texttt{turnRight()} \mid \texttt{turnLeft()} \\
&\quad \mid \quad \texttt{pickMarker()} \mid \texttt{putMarker()} \\
\text{Cste } r \quad &:= \quad 0 \mid 1 \mid \cdots \mid 19
\end{aligned}
$$

Figure 8: The Karel domain-specific language (DSL) (Bunel et al., 2018).

## C.5 BASICKAREL

This environment, introduced by Tzannetos et al. (2023), is a variant of KAREL that excludes control flow constructs such as loops and conditionals, and only includes basic actions. We provide the details here for completeness.

The environment consists of an agent inside a $4 \times 4$ grid. Each cell in the grid could contain a marker, correspond to a wall, or be empty. The objective of each task is to generate a sequence of actions that transforms a pre-grid to a post-grid. The BASICKAREL dataset has 24000 training tasks and 2400 validation tasks. The set of initial states, $\mathcal{S}_{\text{init}}$, is the same as the training set of tasks provided in the BASICKAREL dataset. Each state $s \in \mathcal{S}$ corresponds to a tuple $(\texttt{Curr-Grid}, \texttt{Post-Grid})$, where `Curr-Grid` and `Post-Grid` correspond to the bitmap representation of the current-grid and the post-grid, respectively.

The action space is $\mathcal{A} = \{\texttt{move}, \texttt{turnLeft}, \texttt{turnRight}, \texttt{pickMarker}, \texttt{putMarker}, \texttt{finish}\}$. `move` moves the agent in the facing direction. `turnLeft` and `turnRight` turn the agent left and right, respectively. The agent can pick and put a marker using `pickMarker` and `putMarker`, respectively. The agent gets a reward of 1 on executing `finish` if `Curr-Grid` matches `Post-Grid` (i.e., it has successfully transformed the pre-grid to the post-grid); `finish` results in a crash otherwise. The horizon length is 20 and $\gamma$ is 0.999.

## D  TECHNICAL DETAILS OF THE PREDMODEL BASELINE

**Method.** We begin by constructing a dataset $\mathcal{D}$ comprising transitions $(s_0, \bar{s}_t, a_t, r_{t+1}, \bar{s}_{t+1})$, obtained from $\text{N}_\text{R}$ rollouts (with each rollout comprising multiple transitions) of the expert multi-task policy in the MDP from randomly sampled tasks, where $s_0$ represents a task, and $\bar{s}$ denotes state $s$ with context variables removed. The task embedding $z$ is inferred through variational inference.

More concretely, we train an inference network $q_\phi(z \mid s_0)$, parameterized by $\phi$, which is modeled as a diagonal Gaussian approximate posterior over z. This network is trained to reconstruct the MDP through predictive models of reward $f_{\theta_r}^{(r)}$ and dynamics $f_{\theta_s}^{(s)}$, both parameterized by $\theta_r$ and $\theta_s$, respectively. Essentially, we solve the following optimization problem:

$$\min_{\phi, \theta_r, \theta_s} \left[ \mathbb{E}_{(s_0, \bar{s}_t, a_t, r_{t+1}, \bar{s}_{t+1}) \sim \mathcal{D}} \left[ \beta D_{\mathrm{KL}}(q_\phi(z|s_0) \| p(z)) \right. \right.$$

$$\left. \left. + \mathbb{E}_{z \sim q_\phi(z|s_0)} \left[ \alpha_r \| f_{\theta_r}^{(r)}(\bar{s}_t, a_t, z) - r_{t+1} \|_2^2 + \alpha_s \| f_{\theta_s}^{(s)}(\bar{s}_t, a_t, z) - s_{t+1} \|_2^2 \right] \right] \right],$$

where $p(z)$ is a standard normal prior over $z$, and $\alpha_r$, $\alpha_s$, and $\beta$ are hyperparameters.

**Implementation details.** For each environment, $N_R$ is set to 10000, and the inference network, as well as the predictive models, are implemented as feedforward neural networks with 2 hidden layers (128 neurons in each layer) and ReLU activations. The predictive models share weights, except for the final layer. We set $\alpha_r$, $\alpha_s$, and $\beta$, to 1, 1, and 0.01, respectively. The networks are jointly trained for 500 epochs using the Adam optimizer with 0.001 learning rate and a batch size of 512. The embedding dimensionalities are set to 6, 3, 3, and 8, for MULTIKEYNAV, CARTPOLEVAR, POINTMASS, and BASICKAREL, respectively.

## E  IMPLEMENTATION DETAILS

**Compute resources.** All the experiments were conducted on a cluster of machines with Intel Xeon Gold 6134M CPUs (clocked at 3.20 Ghz) and Nvidia Tesla V100 GPUs (32 GB VRAM configuration). We would like to highlight that learning the task embedding function is a one-time process, which could be compute intensive. However, once the training is completed, computing the embeddings or utilizing them to measure task similarity is a one-shot operation.

**Training process.** For all the environments, we set $\delta_{\mathrm{snap}}$ to 0.01 (0.1 for KAREL) in our experiments. The average performance on $\mathcal{S}_{\mathrm{snap}}$ is determined by performing 10 rollouts on each task in the set. We implement each embedding network using a succession of several fully connected layers with ReLU activations, trained for 300 epochs (500 epochs for CARTPOLEVAR and 10 epochs for KAREL) using the Adam optimizer with $1\mathrm{e}-3$ ($1\mathrm{e}-4$ for KAREL) learning rate, and batch size 128 (512 for KAREL). We sample 5000 (80000 for KAREL and 10000 for BASICKAREL) constraints from $\mathcal{C}_{\mathrm{MI}}$ and $\mathcal{C}_{\mathrm{NORM}}$ each to train the network, the $\hat{\mathcal{I}}$ values of which are approximated using 100 samples from each agent in the population. p_success(.) is approximated using 10 samples from each agent in the population. The hyperparameters $\beta$ and $\lambda$ are set to 1000 and 0.4, respectively. In addition, we use a validation set and a test set, consisting of 1000 (16000 for KAREL and 2000 for BASICKAREL) constraints from $\mathcal{C}_{\mathrm{MI}}$ and $\mathcal{C}_{\mathrm{NORM}}$ each, for early stopping and to determine the final model parameters. We vary the embedding dimensionality from 1 to 10 and choose the one after which the test loss does not decrease much. Below we provide environment-specific details:

- **MULTIKEYNAV**: $\mathcal{S}_{\mathrm{snap}}$ includes all combinations of the locations $\{0.05, 0.45, 0.85\}$, key statuses, and door types. We combine the subpopulations obtained by masking no action, masking each `pickKey` action individually, and masking all `pickKey` actions, to obtain an agent population of size 100. The embedding network has two hidden layers (32 neurons in each layer). The embedding dimensionality is 5 without $\mathcal{C}_{\mathrm{NORM}}$, and 6 with $\mathcal{C}_{\mathrm{NORM}}$.

- **CARTPOLEVAR**: $\mathcal{S}_{\mathrm{snap}}$ consists of 1000 tasks sampled from $\mathcal{S}_{\mathrm{init}}$. We combine the subpopulations obtained by using all tasks in $\mathcal{S}_{\mathrm{init}}$, tasks in $\mathcal{S}_{\mathrm{init}}$ with +ve F and Type 0, tasks in $\mathcal{S}_{\mathrm{init}}$ with +ve F and Type 1, tasks in $\mathcal{S}_{\mathrm{init}}$ with $-$ve F and Type 0, and tasks in $\mathcal{S}_{\mathrm{init}}$ with $-$ve F and Type 1, to obtain an agent population of size 95. The embedding network has two hidden layers (64 neurons in the first layer and 32 neurons in the second). The embedding dimensionality is 2 without $\mathcal{C}_{\mathrm{NORM}}$, and 3 with $\mathcal{C}_{\mathrm{NORM}}$.

- **POINTMASS**: $\mathcal{S}_{\mathrm{snap}}$ consists of 100 tasks sampled from $\mathcal{S}_{\mathrm{init}}$. We combine subpopulations obtained by using all the tasks in $\mathcal{S}_{\mathrm{init}}$, tasks in $\mathcal{S}_{\mathrm{init}}$ that satisfy the condition $\mathtt{p_g} + 0.5\mathtt{w_g} < 0$, and tasks in $\mathcal{S}_{\mathrm{init}}$ that satisfy the condition $\mathtt{p_g} + 0.5\mathtt{w_g} \geq 0$, to obtain an agent population of size about 25. The embedding network has two hidden layers (32 neurons in each layer). The embedding dimensionality is 3 with $\mathcal{C}_{\mathrm{NORM}}$.

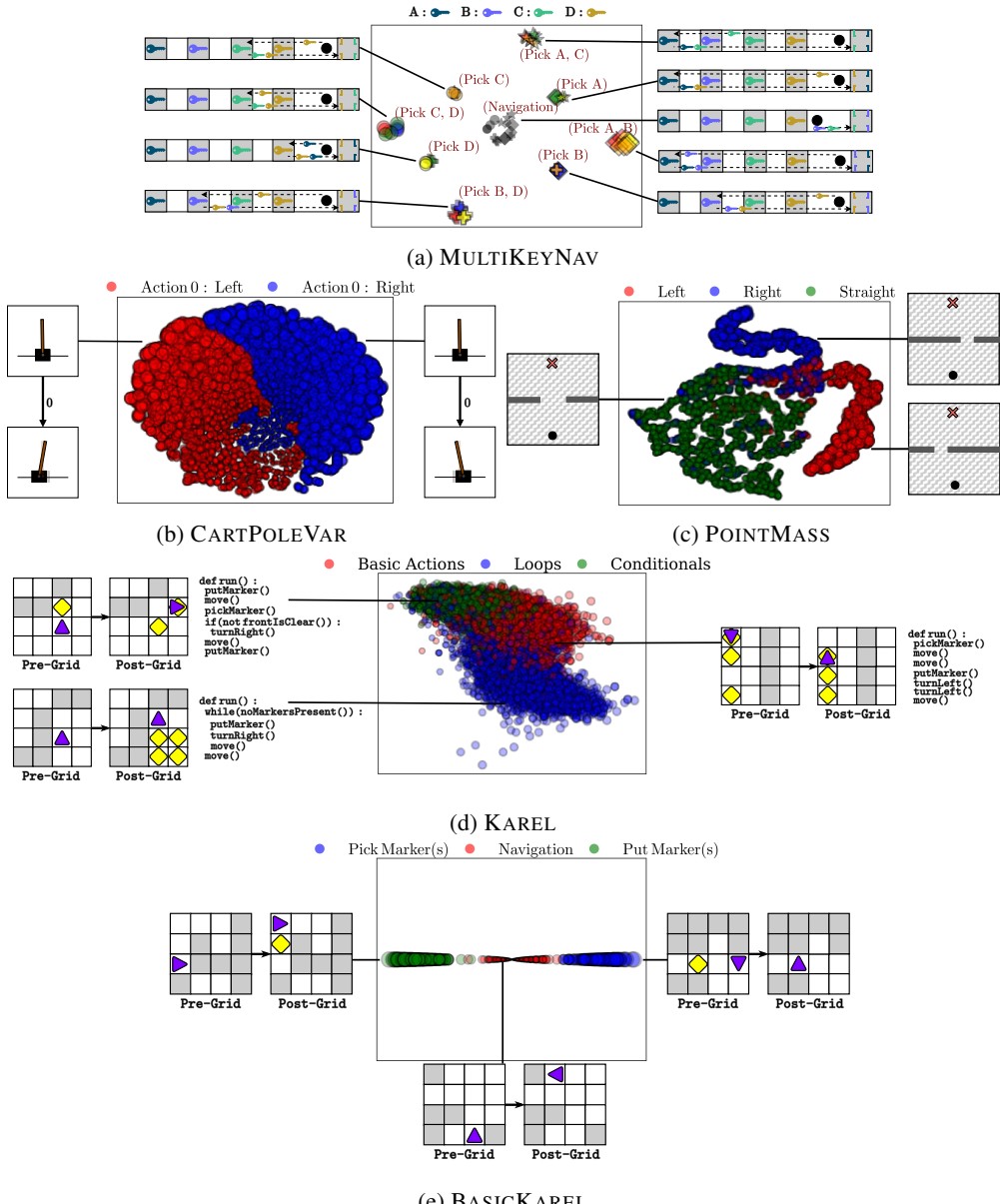

Figure 9: Annotated visualization of the task embedding spaces with example tasks for each cluster.

- **KAREL**: $\mathcal{S}_{\mathrm{snap}}$ consists of 14681 tasks sampled from the Karel dataset. We combine subpopulations obtained by using all the tasks in $\mathcal{S}_{\mathrm{init}}$, tasks in $\mathcal{S}_{\mathrm{init}}$ that do not require synthesizing token for loops, tasks in $\mathcal{S}_{\mathrm{init}}$ that do not require synthesizing tokens for conditionals, and tasks in $\mathcal{S}_{\mathrm{init}}$ that do not require synthesizing tokens for both loops and conditionals, to obtain an agent population of size about 135. We use the official codebase of (Bunel et al., 2018) to train the agents. The embedding network consists of an input-output encoder (which is the same as that in (Bunel et al., 2018)) followed by a feedforward network with a single hidden layer (256 neurons). The embedding dimensionality is 2 with $\mathcal{C}_{\mathrm{NORM}}$.

- **BASICKAREL**: $\mathcal{S}_{\mathrm{snap}}$ consists of all the 2400 validation tasks. We combine the subpopulations obtained by masking no action, masking `pickMarker`, masking `putMarker`, and masking both `pickMarker` and `putMarker`, to obtain an agent population of size about 55. The embedding network has two hidden layers (32 neurons in each layer). The embedding dimensionality is 1 with $\mathcal{C}_{\mathrm{NORM}}$.

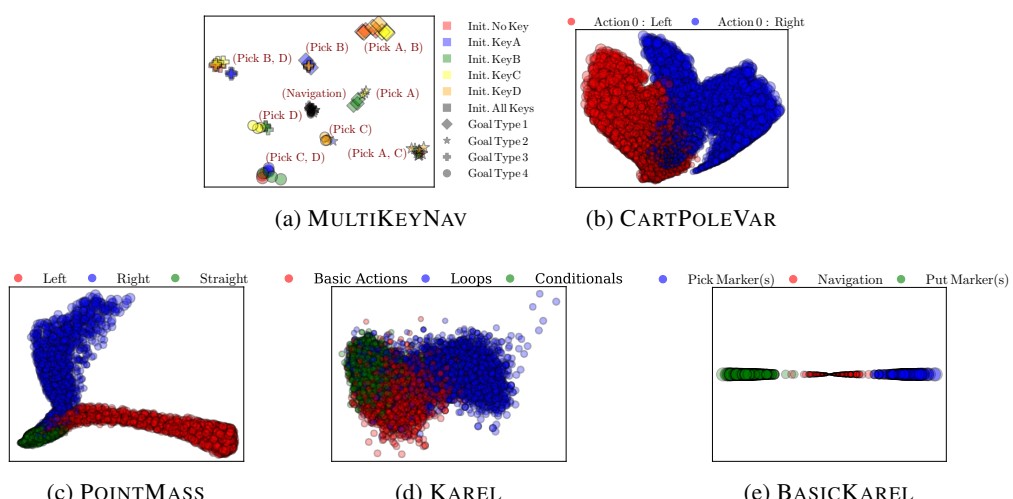

Figure 10: PCA projections of the task embedding spaces learned through our framework. Each point represents a task, and the size of the points is proportional to the norm of the embeddings.

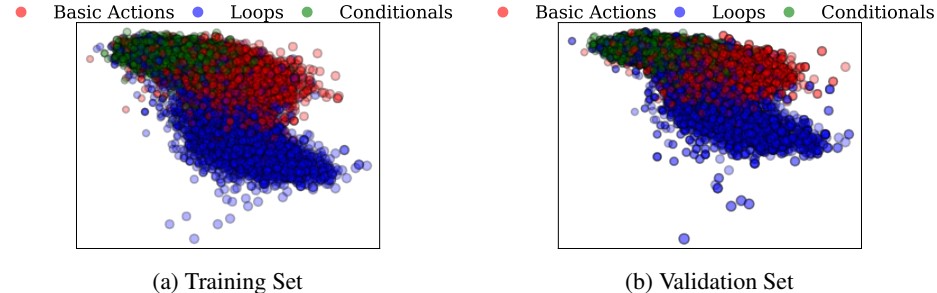

Figure 11: Task embedding spaces obtained using the training and validation sets of tasks for the KAREL environment. This visualization shows that an embedding function learned using a training set of tasks generalizes to a validation set of tasks.

**Performance prediction benchmark.** We use 500 samples to estimate the *IgnoreTask* baseline, 10 samples from each agent in the population to estimate the *IgnoreAgent* baseline, and 10 samples to estimate the *OPT* baseline. $\beta$ is tuned using the training examples.

**Task selection benchmark.** We use 100 samples from each agent in the population to estimate task similarity, and 10 samples from each agent to estimate task difficulty.

## F   ANNOTATED VISUALIZATION OF EMBEDDING SPACES

In Fig. 9, we visualize the learned embedding spaces annotated with an example task for each cluster.

## G   PCA PLOTS

Fig. 10 visualizes the learned task embedding spaces through 2D PCA projections.

## H   GENERALIZATION EXPERIMENT

We present a generalization experiment in which we assess if the embedding function (learned using a training set of tasks) produces a consistent embedding space for a validation set of tasks. We conduct this evaluation on the KAREL environment by partitioning the tasks into training and validation sets of size 59007 and 14681, respectively. Fig. 11 visualizes the task embedding spaces for these sets and shows the generalization ability of the learned task embedding function.

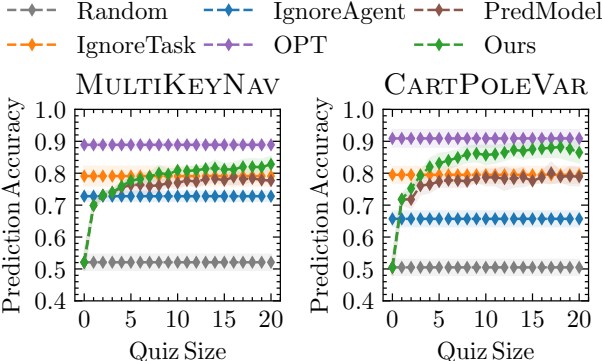

Figure 12: Performance prediction results for new agents using task embeddings. Our technique is competitive with the *OPT* baseline, which is the best one could do on this benchmark.

# I   PERFORMANCE PREDICTION – NEW AGENTS

To assess if the correlations captured by our similarity criterion remain valid for new agents, we evaluate the learned task embedding network in the performance prediction application scenario for new agents. For MULTIKEYNAV, the new population of agents is created using biased task distributions instead of action masking. More concretely, we combine the subpopulations obtained by using all tasks, tasks with doors of Type 1, tasks with doors of Type 2, tasks with doors of Type 3, and tasks with doors of Type 4. For CARTPOLEVAR, we use the Proximal Policy Optimization (PPO) algorithm (Schulman et al., 2017) instead of behavioral cloning to create the new agent population.

Fig. 12 shows the prediction accuracies of various techniques. Our method is competitive with *OPT*, demonstrating the efficacy of task embeddings in predicting the performance of new agents.

