# OpenReview forum: "Learning Embeddings for Sequential Tasks Using Population of Agents"
_ICLR.cc/2024/Conference — Submitted to ICLR 2024_

### Official Review · Reviewer_9bsB · 2023-10-29

**Soundness:** 2 fair
**Presentation:** 3 good
**Contribution:** 2 fair
**Rating:** 5
**Confidence:** 3

**Summary:**

The authors propose a framework to learn fixed dimensional task embeddings for RL tasks. Their goal is to ensure that tasks with similar embeddings have similar performance across a diverse population of agents. The similarity measure used is information theoretically motivated and the authors propose an algorithm to learn the task embeddings satisfying ordinal constraints imposed by this similarity measure. The learned embeddings are visually demonstrated for 5 tasks: MULTIKEYNAV, CARTPOLEVAR, POINTMASS, KAREL and BASICKAREL. Finally, quantitative results are provided showing the effectiveness of the learned embeddings in predicting performance on similar tasks and for identifying tasks with desired characteristics in the MULTIKEYNAV and CARTPOLEVAR settings.

**Strengths:**

1. The proposed framework is intuitive and easy to follow. The writing overall is also easy to understand.

2. Using learned task embeddings to reduce uncertainty about agent's performance on unseen tasks based on its performance on related tasks could be helpful in different RL applications, therefore the problem setup seems to be well-motivated.

3. For the 5 environments considered in the paper, extensive experiments have been performed to analyze the performance of the proposed method.

**Weaknesses:**

1. The results included in the paper focus on learning low dimensional embeddings for the tasks - for example, in CARTPOLEVAR the learnt embedding is of dimension 2 or 3 whereas in BASICKAREL it is of dimension 1. The experiments do not consider more difficult tasks, such as MuJoCo tasks considered in [1].

2.  There is no discussion of the relatedness / differences with the bisimulation representation learning method in [1] which also learns an embedding of states in RL tasks, and ensures that states which would lead to similar outcomes have similar embeddings. It would help to include a discussion of why it has not been considered as a baseline in the experiments either.

3. It is a bit confusing to understand the differences between $S_{init}$ and $S$. The authors should consider clarifying in the main paper the differences between a task definition and the MDP states.

4. The proposed method relies heavily on the availability of a diverse set of agents in the environment. This could affect the quality of task embeddings learned, as the authors also demonstrate in Fig. 4.

[1] Zhang, A., McAllister, R., Calandra, R., Gal, Y. and Levine, S., 2020. Learning invariant representations for reinforcement learning without reconstruction. arXiv preprint arXiv:2006.10742.

**Questions:**

I do not fully understand the PredModel baseline. The authors say it is "inspired by prior work" but there are no citations provided and I may be missing the link to prior work. Could the authors please clarify that?

---

> ### Author Response · Authors · 2023-11-16
>
> We thank the reviewer for carefully reviewing our paper! We greatly appreciate the feedback. Please see below our responses to the comments.
>
> -----
> **… differences between a task definition and the MDP states …**
>
> We would like to provide some intuition by connecting our formalism of tasks as initial states to the contextual MDP setting [Hallak et al., 2015], where each context (e.g., MultiKeyNav's context: agent's initial position, possessed keys initially, door type) corresponds to a distinct task represented by a separate MDP with context-dependent transitions and rewards. This set of MDPs can be converted into an equivalent MDP by including context variables as part of the state. In this converted MDP, each initial state corresponds to a task, as it determines the context for the entire episode. The context is observable.
>
> -----
> **… relatedness / differences with the bisimulation representation learning method in [1]**
>
> Our work differs significantly from [1] and other state representation learning methods, as our focus is on learning representations for tasks rather than individual states. Although our definition of tasks as initial states might indicate an interest in state representations, our formalism is connected to the contextual MDP setting and we are interested in learning representations for context. Proximity in the context space doesn't necessarily imply task similarity. This is where our work makes its contribution by mapping the context to task embeddings. We will include a discussion on this in the camera-ready version of the paper.
>
> -----
> **The experiments do not consider more difficult tasks, such as MuJoCo tasks considered in [1].**
>
> The environments used in [1] are designed to evaluate robustness to task-irrelevant distractors. These distractors, however, do not introduce any variability in the reward function or transition dynamics. This choice is perhaps deliberate, as the objective is to compare methods for learning state representations. In contrast, we are interested in learning representations for tasks, where variability occurs in the reward function and/or dynamics.
>
> In summary, we use MultiKeyNav as the running example because it is compositional (the agent needs to compose different actions for picking keys in a task-specific manner to unlock the door), which also makes it suitable for ablation experiments. Considering that task variability in MultiKeyNav comes from the reward function, we use CartPoleVar to highlight our framework’s applicability to environments in which task variability comes from the dynamics instead. We select PointMass to test if our framework can handle continuous action spaces. Finally, to investigate our framework’s scalability, we use the real-world environment Karel, which is a challenging environment (with high-dimensional states) that has applications in programming education.
>
> -----
> **I do not fully understand the PredModel baseline. … there are no citations provided …**
>
> PredModel is inspired by PEARL [Rakelly et al., 2019]. We discuss the technical details of this baseline in Appendix D.
>
> -----
>
> [Hallak et al., 2015] Contextual Markov Decision Processes. arXiv:1502.02259 2015.
>
> [Rakelly et al., 2019] Efficient Off-Policy Meta-Reinforcement Learning via Probabilistic Context Variables. ICML 2019.
>
> -----
>
> We hope that our responses address your concerns and are helpful in improving your rating. If you have any other comments or feedback, please let us know! We are looking forward to hearing back from you! Thank you again for the review.

---

> > ### Comment · Reviewer_9bsB · 2023-11-21
> > **Acknowledging authors' response**
> >
> > Thank you for the response and the clarifications about the problem setup.
> >
> > >  Although our definition of tasks as initial states might indicate an interest in state representations, our formalism is connected to the contextual MDP setting and we are interested in learning representations for context.
> >
> > I am not fully convinced by this argument that framing the tasks in the contextual MDP setup (but then reducing context to the initial state space) makes it sufficiently different to not merit comparison with state representation learning methods.
> >
> > > The environments used in [1] are designed to evaluate robustness to task-irrelevant distractors.
> >
> > Yes, and perhaps the proposed method in this paper should also be evaluated for robustness to distractors, since in practical applications, a clean context or task definition may not always be available.
> >
> >
> > Therefore I will maintain my current rating for the paper in its current form.

---

> ### Author Response · Authors · 2023-11-21
>
> Dear Reviewer 9bsB,
>
> Since the Author/Reviewer Discussion period concludes on Nov. 22, we would be glad to address any remaining concerns or questions.
>
> Thanks,\
> Authors

---

### Official Review · Reviewer_HkeS · 2023-10-31

**Soundness:** 2 fair
**Presentation:** 3 good
**Contribution:** 2 fair
**Rating:** 5
**Confidence:** 4

**Summary:**

This paper studies the problem of learning embeddings for RL tasks that capture the semantics of these tasks. In particular, the goal is to represent tasks using finite dimensional vectors such that (i) the dot product of the vectors corresponding to any two tasks measures the similarity between the tasks and (ii) the norm of the vector representing a task captures the difficulty of the task. The solution involves quantifying task similarity and task difficulty using a distribution of diverse agents and then learning embeddings to optimize the two objectives. Experiments on different environments show that the learned embeddings indeed satisfy the two objectives—e.g., they can be used to obtain clusters of similar tasks. The usefulness of such embeddings is demonstrated by using them to solve two downstream tasks: (i) predicting the performance of a policy w.r.t. a task given its performance on a small set of tasks and (ii) selecting a task from a set of tasks that satisfies various criteria (such as most similar to a given task).

**Strengths:**

- The idea of learning general purpose embeddings for tasks instead of learning them for the specific purpose of multi-task learning seems novel and interesting. The studied applications (performance prediction and task selection) justify the value in learning such embeddings. These applications can be useful in other domains such as curriculum learning.
- Using a distribution over a diverse population of agents to quantify difficult-to-express objectives such as task similarity is an interesting technique and can potentially be applied in other scenarios.
- The paper is fairly well-written and conveys the main ideas clearly (though some details could be explained better).

**Weaknesses:**

- The entire approach seems to depend heavily on the population of agents used to define the learning objectives. For instance, the probability of success (PoS) of a task is taken to be a measure of task difficulty. However, it is possible that an “easy” task has a lower PoS when compared to a “difficult” task if a policy solving the easy task is absent in the set of agents. Some of the experimental results seem to be a direct result of the way the agent population is obtained—e.g., the clusters corresponding to unique sets of keys in MultiKeyNav could be a result of using biased task distributions to train agents. Some heuristics are suggested for obtaining the population of agents which seem to work well for the environments in the paper, but their applicability to new domains is unclear.
- The overall task is assumed to be representable by the initial state. This enables task embedding to be a function of the initial state. This assumption might not hold in general (several tasks could start from the same state and vice-versa). In such cases, the task is represented by the reward function and the proposed approach is not readily applicable.
- Some comparisons to baselines seem unfair since the evaluation criterion is based on the population of agents used to learn the embeddings. For instance, in the task selection experiment, task similarity and difficulty (for evaluation purposes) are measured using the same quantities as those used while learning the embeddings. Therefore, the significance of these experiments is unclear.

**Questions:**

1. In Section 5.4, PredModel is mentioned to be inspired by prior work. Could you provide a citation for the work this baseline is inspired by?
1. Why is the start state assumed to represent the task and why is it a reasonable assumption? Are there other ways to represent tasks (so that they can be input to the embedding network) such as natural language descriptions that are better suited here?
1. It looks like transfer learning and multi-task learning are natural applications of such embedding vectors. Are the generated embeddings helpful for these applications as well?

---

> ### Author Response · Authors · 2023-11-16
> **Response to Reviewer HkeS - Part 1**
>
> We thank the reviewer for carefully reviewing our paper! We greatly appreciate the feedback. Please see below our responses to the comments.
>
> -----
> **The entire approach seems to depend heavily on the population of agents … it is possible that an “easy” task has a lower PoS when compared to a “difficult” task if a policy solving the easy task is absent in the set of agents.**
>
> Indeed, the agent population serves as an important component in our framework. Essentially, we gather task correlation statistics from an agent population that emerges during the multi-task policy learning process. The scenario where “a policy solving an easy task is absent from the agent population” highlights the significance of having a good-quality population. For instance, please refer to our ablation experiment w.r.t. the population specification in Section 5.3, where agents in the population fail to learn to pick keys. This experiment shows how a poor-quality population affects both our proposed similarity criterion and the resulting task embedding space (see Fig. 4b). Thus, the population of agents must be created by appropriately choosing a good learning algorithm and a diversity mechanism. A promising research direction is to develop more principled approaches for generating this population, possibly optimizing for specific diversity measures.
>
> -----
> **the clusters corresponding to unique sets of keys in MultiKeyNav could be a result of using biased task distributions to train agents. Some heuristics are suggested for obtaining the population of agents … but their applicability to new domains is unclear.**
>
> Please refer to Appendix E for more details on how the agent populations are created. We utilize action masking for MultiKeyNav, where the effective task is determined by the door type (dictating required keys) and the agent's initial key possession (determining remaining keys to pick up). Thus, the task embedding space for this environment is not merely an outcome of action masking; the embedding network needs to reason about the relationship between door types and initial keys for determining the effective task.
>
> We agree that the heuristics for creating a diverse population of agents might require some domain knowledge. Therefore, a promising research direction is to develop more principled approaches for generating this population, possibly optimizing for specific environment-agnostic diversity measures.
>
> -----
> **The overall task is assumed to be representable by the initial state. … This assumption might not hold in general (several tasks could start from the same state and vice-versa). In such cases, the task is represented by the reward function and the proposed approach is not readily applicable.**
>
> Our formalism of tasks as initial states is connected to the contextual MDP setting [Hallak et al., 2015], where each context (e.g., MultiKeyNav's context: agent's initial position, possessed keys initially, door type) corresponds to a distinct task represented by a separate MDP with context-dependent transitions and rewards. This set of MDPs can be converted into an equivalent MDP by including context variables as part of the state. In this converted MDP, each initial state corresponds to a task, as it determines the context for the entire episode. The context is observable. Thus, two MDPs that differ in terms of the reward function can be represented within our framework. This scenario is illustrated in MultiKeyNav, where distinct tasks share similar transition dynamics but may have different reward functions.
>
> -----
> (the response is continued in Part 2)

---

> ### Author Response · Authors · 2023-11-16
> **Response to Reviewer HkeS - Part 2**
>
> (continuation of the response from Part 1)
>
> -----
> **Some comparisons to baselines seem unfair since the evaluation criterion is based on the population of agents used to learn the embeddings.**
>
> For Section 6, our baselines correspond to different levels of oracle knowledge. For instance, in the performance prediction benchmark (Section 6.1), OPT represents the oracle that can observe both the agent and the task, and Ignore-Agent represents the one that can observe the task but not the agent. Additionally, in Appendix I, we evaluate the performance of the task embedding networks in the performance prediction application scenario for new agent populations different from those used to train the embedding networks.
>
> For the task selection benchmark (Section 6.2), our baselines consist of various oracles, namely OPT and $\widehat{\text{OPT}}\_\text{50}$, along with alternative one-shot measures of “similarity” between tasks, such as StateSim and TrajectorySim. The purpose of comparing with StateSim and TrajectorySim is to evaluate how our definition of task similarity differs from other one-shot measures. The results indicate that our proposed task similarity criterion captures correlations that extend beyond direct measurements of differences in initial state representations or the edit distance between expert trajectories.
>
> PredModel is included for completeness. However, as we acknowledge in Section 2, our work is positioned as complementary to existing methods.
>
> -----
> **… PredModel is mentioned to be inspired by prior work. Could you provide a citation for the work this baseline is inspired by?**
>
> PredModel is inspired by PEARL [Rakelly et al., 2019].
>
> -----
> **Are there other ways to represent tasks … such as natural language descriptions …**
>
> As discussed previously, our formalism of tasks as initial states is connected to the contextual MDP setting. The context could correspond to natural language descriptions and is covered by our formalism.
>
> -----
> **… transfer learning and multi-task learning … Are the generated embeddings helpful for these applications as well?**
>
> While our framework assumes access to policies that can already solve tasks in a multi-task setup, it would be interesting to explore the possibility of jointly learning task embeddings and multi-task policies using our framework.
>
> -----
>
> [Hallak et al., 2015] Contextual Markov Decision Processes. arXiv:1502.02259 2015.
>
> [Rakelly et al., 2019] Efficient Off-Policy Meta-Reinforcement Learning via Probabilistic Context Variables. ICML 2019.
>
> -----
>
> We hope that our responses address your concerns and are helpful in improving your rating. If you have any other comments or feedback, please let us know! We are looking forward to hearing back from you! Thank you again for the review.

---

> ### Author Response · Authors · 2023-11-21
>
> Dear Reviewer HkeS,
>
> Since the Author/Reviewer Discussion period concludes on Nov. 22, we would be glad to address any remaining concerns or questions.
>
> Thanks,\
> Authors

---

> > ### Comment · Reviewer_HkeS · 2023-11-23
> > **Author Response Acknowledgement**
> >
> > I thank the authors for providing a detailed response. I read the response carefully and I would like to maintain my current score since the main drawbacks mentioned in my review are still present.

---

### Official Review · Reviewer_Q8ok · 2023-11-01

**Soundness:** 2 fair
**Presentation:** 3 good
**Contribution:** 2 fair
**Rating:** 5
**Confidence:** 3

**Summary:**

This paper introduces an algorithm for learning task embeddings to measure the difficulty and similarity between tasks through the performance of a population of agents given a class of tasks. The algorithm includes two components (1) contrast among a triplet of tasks, making sure the inner product of the task embeddings implies the task similarity (2) impose the constraint on the easier tasks that have smaller norms. Experiments test the following hypothesis:

1. Distinct clusters can be visualized through embedding space
2. The norm of the embeddings can indicate task difficulty
3. The learned embedding can be used to predict the agent’s performance and task selection with desired characteristics

**Strengths:**

1. The paper is written in clarity and the logics are easy to follow
2. This paper does nice visualization and the results make sense

**Weaknesses:**

1. It is unclear how task embedding is useful to me. As it requires checkpoints of learned policies that almost solve the task and "difficulty" is vague to an agent's performance as an agent may take a different path to solve the task when there are multiple solutions. Plus, it is almost impossible to get task embedding without exploring a few trajectories of it to get anything meaningful, unlike some rule-description tasks.
2. It only generalizes to variations of a particular environment.

**Questions:**

1. How do you guarantee the diversity of the population?
2. Did you test learning task embedding using a single agent?
3. How the *agent performance* data is collected? What agents did you use? Were they involved in the training of the embedding?
4. How to test the generalization of the task embedding? (aka generalize across different tasks.)

---

> ### Author Response · Authors · 2023-11-16
>
> We thank the reviewer for carefully reviewing our paper! We greatly appreciate the feedback. Please see below our responses to the comments.
>
> -----
> **It is unclear how task embedding is useful to me. As it requires checkpoints of learned policies that almost solve the task and "difficulty" is vague to an agent's performance as an agent may take a different path to solve the task when there are multiple solutions.**
>
> Our framework utilizes an agent population that emerges during the multi-task policy learning process to collect task correlation statistics and estimate task difficulty. The proposed task similarity criterion is based on the collective behavior of the agent population, while task difficulty, defined as the probability of a random agent from the population failing on the task, is an aggregate statistic computed for the entire agent population, not individually per agent. Thus, these quantities are population-level statistics.
>
> The task embedding network learned through our framework offers the following advantages:
>
> 1. It enables one-shot computation of task similarity without requiring experience data or access to the agent population.
> 2. It can generalize to new tasks, as demonstrated in Appendix H. More concretely, we show that it generalizes from a training set to a validation set of tasks not encountered during agent population training or embedding network learning.
> 3. Our proposed task similarity criterion, which is used to train the embedding network, captures performance correlations that remain valid for new agents. We demonstrate this in Appendix I by assessing the performance of the task embedding networks in the performance prediction application scenario for different agent populations.
>
> -----
> **It only generalizes to variations of a particular environment.**
>
> Our formalism of tasks is connected to the contextual MDP setting [Hallak et al., 2015], where each context (e.g., MultiKeyNav's context: agent's initial position, possessed keys initially, door type) corresponds to a distinct task represented by a separate MDP with context-dependent transitions and rewards. It's important to note that proximity in the context space doesn't necessarily imply task similarity. This is where our work makes its contribution by mapping the context to task embeddings. However, we must note that our framework is general enough to cover the scenario where tasks correspond to different environments. The challenge here lies in learning multi-task policies that can jointly solve these environments.
>
> -----
> **How do you guarantee the diversity of the population?**
>
> We introduce action masking or biased task distributions, as ways to diversify the agent population. A promising research direction is to develop more principled approaches for generating this population, possibly optimizing for specific diversity measures.
>
> -----
> **Did you test learning task embedding using a single agent?**
>
> As previously mentioned, our proposed task similarity criterion is formulated as a population-level statistic. When the agent population is reduced to a singleton, it evaluates to zero by definition, which effectively makes the criterion uninformative.
>
> -----
> **How the agent performance data is collected? What agents did you use? Were they involved in the training of the embedding?**
>
> The benchmarks for the agent performance prediction scenario in Section 6.1 were created with the same agent population used to learn the task embedding network. However, our proposed task similarity criterion captures performance correlations that remain valid for new agents. We demonstrate this in Appendix I by assessing the performance of the task embedding network in the performance prediction application scenario for different agent populations.
>
> -----
> **How to test the generalization of the task embedding? (aka generalize across different tasks.)**
>
> The learned task embedding network can generalize to new tasks, as demonstrated in Appendix H. More concretely, we show that it generalizes from a training set to a validation set of tasks not encountered during agent population training or embedding network learning.
>
> -----
>
> [Hallak et al., 2015] Contextual Markov Decision Processes. arXiv:1502.02259 2015.
>
> -----
>
> We hope that our responses address your concerns and are helpful in improving your rating. If you have any other comments or feedback, please let us know! We are looking forward to hearing back from you! Thank you again for the review.

---

> ### Author Response · Authors · 2023-11-21
>
> Dear Reviewer Q8ok,
>
> Since the Author/Reviewer Discussion period concludes on Nov. 22, we would be glad to address any remaining concerns or questions.
>
> Thanks,\
> Authors

---

> ### Comment · Reviewer_Q8ok · 2023-11-21
> **Acknowledge of reading the author response**
>
> Dear authors,
>
> Thank you for providing the response to address my questions! I have to appreciate that I can see the great effort of the authors in the paper and in the rebuttal phase.
>
> I read through other reviews and responses as well to try to look for answers to some of my questions, I think all the reviewers share the following three concerns:
>
> - (1) Motivation of the task embedding.
> - (2) Diversity guarantee of the population. (Is "hard" really hard, and "easy" really easy)
> - (3) Generalization across environments. (I think I am clear about it at this point that the "task" is defined within an environment so I am not going to be harsh with this.)
>
> For the motivation, I am wondering if the authors could further elaborate on how "task embedding" can actually help with "what". For example, are you using it to improve the curriculum learning task selection or improve other aspects of any algorithms? Does it help with any efficiency of algorithms or is it a solid evaluation or estimation methodology, or other applications? I would like to see that statement as the current explanation of framework advantage is not quite convincing yet. For example, "one-shot computation of task similarity", yes, and does it imply "quicker adaptation of the policy to other tasks"?
>
> I did see another paper https://arxiv.org/abs/2310.07218 that also adopts populations and information-theoretic metrics working on different topics, but their motivation is clearer to me -- estimate the need for multi-agent interactions through a less computationally expensive method (selfplay) to decide whether we need a more complex algorithm (fictitiou co-play https://arxiv.org/abs/2110.08176).

---

> > ### Author Response · Authors · 2023-11-22
> > **Follow-up Response to Reviewer Q8ok**
> >
> > We thank the reviewer for their response. Please see below our responses to the questions.
> >
> > -----
> > **Motivation of the task embedding**
> >
> > Several methods for curriculum design in RL rely on the agent's value function (e.g., [Klink et al., 2020], [Eimer et al., 2021], and [Yengera et al., 2021]). Thus, one concrete application of task embeddings is in black-box curriculum design, where only the agent's performance is observable, not the value function. This setting is of practical relevance, considering that black-box agents can be humans. As demonstrated in the performance prediction application scenario (Section 6.1), task embeddings can be used to infer an agent's probability of success on a task by observing its performance on a small quiz. This estimated probability of success serves as an approximation for the agent's value function when the reward is binary and sparse.
> >
> > We plan to evaluate this downstream application in future work.
> >
> > -----
> > **Is "hard" really hard, and "easy" really easy**
> >
> > The definition of task difficulty in our framework is grounded in statistical measures derived from the population of agents.  It is not based on the subjective notion of “hardness” or “easiness”.
> >
> > -----
> >
> > [Klink et al., 2020] Self-Paced Deep Reinforcement Learning. NeurIPS 2020.
> >
> > [Eimer et al., 2021] Self-Paced Context Evaluation for Contextual Reinforcement Learning. ICML 2021.
> >
> > [Yengera et al., 2021] Curriculum Design for Teaching via Demonstrations: Theory and Applications. NeurIPS 2021.
> >
> > -----
> >
> > Please let us know if you have additional comments or feedback. Thanks!

---

### Official Review · Reviewer_viG4 · 2023-11-07

**Soundness:** 3 good
**Presentation:** 3 good
**Contribution:** 3 good
**Rating:** 6
**Confidence:** 3

**Summary:**

The authors develop a framework for comparing task similarity in goal-conditioned settings under a given population of agents. Under this embedding, the norm describes task difficulty and the inner product encodes a notion of similarity.

They perform experiments on CartPoleVar, MultiKeyNav, PointMass and Karel, demonstrating via t-SNE plots that the embeddings correspond
to salient features of the task. They then demonstrate the application of these task embeddings to predicting task performance and task selection. For task selection the authors consider two types of query, one for selecting the most similar task, and one for selecting the task that is most similar, but more difficult than a given task.

**Strengths:**

* The described framework is well-presented and easy to follow. The properties encoded in the task embeddings are logical.
* The paper is overall well-written and easy to follow
* The application results demonstrate convincing performance improvements over relevant baselines and therefore that the embeddings
learned are meaningful encodings of the task.

**Weaknesses:**

* My major issue with the paper surrounds motivation. Creating this task embedding requires a diverse population of agents which together are
competent on a broad range of the tasks. This is a vast amount of compute relative to the amount required to solve an individual task or even a
reasonably broad range of tasks in the space of tasks. It's therefore not entirely clear to me when such a task embedding would be appropriate. The authors go some way to answering this by demonstrating the usefulness of the embeddings in task prediction and task similarity identification. However, it's not clear to me when either of these tasks would be useful compared to training a single agent on a broader task distribution for the same total compute time required to train the population. However, I think judging future usefulness and method relevance is very difficult and so do not weight this point too strongly.
* Because of the large amount of compute required to build these embeddings, the tasks considered are relatively simple. It would be interesting to consider more complex and higher dimensional tasks, such as by embedding levels in ProcGen.

**Questions:**

* How much compute is required to generate the population of agents and embeddings for the tasks? I could not find this information, although I may have missed it.
* How much variation is there in the embeddings with the population? If I train the population in a different way, can the performance prediction generalise to a different population? For example, can the embedding of a task be used to predict the task performance of an agent trained with a different algorithm?

---

> ### Author Response · Authors · 2023-11-16
>
> We thank the reviewer for carefully reviewing our paper! We greatly appreciate the feedback. Please see below our responses to the comments.
>
> -----
> **My major issue with the paper surrounds motivation. … it's not clear to me when either of these tasks would be useful compared to training a single agent on a broader task distribution for the same total compute time required to train the population.**
>
> We agree that the process of learning the embedding network could be compute-intensive for certain environments. However, post-learning, this network offers the following advantages:
>
> 1. It enables one-shot computation of task similarity without requiring experience data or access to the agent population.
> 2. It can generalize to new tasks, as demonstrated in Appendix H. More concretely, we show that it generalizes from a training set to a validation set of tasks not encountered during agent population training or embedding network learning.
> 3. Our proposed task similarity criterion, which is used to train the embedding network, captures performance correlations that remain valid for new agents. We demonstrate this in Appendix I by assessing the performance of the task embedding networks in the performance prediction application scenario for different agent populations.
>
> The two downstream scenarios presented in our work are inspired by real-world applications; the performance prediction scenario is similar to assessing a student’s proficiency in adaptive learning platforms via a compact quiz [He-Yueya et al., 2021], and the task selection scenario is analogous to selecting desired questions from a pool for a personalized learning experience in online education systems [Ghosh et al., 2022].
>
> -----
> **Because of the large amount of compute required … the tasks considered are relatively simple.**
>
> We selected environments with apparent variability along reward and/or dynamics axes. In summary, we use MultiKeyNav as the running example because it is compositional (the agent needs to compose different actions for picking keys in a task-specific manner to unlock the door), which also makes it suitable for ablation experiments. Considering that task variability in MultiKeyNav comes from the reward function, we use CartPoleVar to highlight our framework’s applicability to environments in which task variability comes from the dynamics instead. We select PointMass to test if our framework can handle continuous action spaces. Finally, to investigate our framework’s scalability, we use the real-world environment Karel, which is a challenging environment (with high-dimensional states) that has applications in programming education.
>
> -----
> **How much compute is required to generate the population of agents and embeddings for the tasks?**
>
> Below we report the approximate run times (in hours) to train a single agent, an agent population, and the embedding network for the environments considered in this work. All the experiments were conducted with an Intel Xeon Gold 6134M CPU (clocked at 3.20 Ghz) and an Nvidia Tesla V100 GPU (32 GB VRAM configuration).
>
>
> | Environment | Single Agent | Agent Population | Embedding Network |
> | ----- | --- | --- | --- |
> | MultiKeyNav | $0.5$ | $3$ | $18$ |
> | CartPoleVar | $0.5$ | $3$ | $1$ |
> | PointMass | $2.5$ | $7.5$ | $40$ |
> | Karel | $20$ | $65$ | $240$ |
> | BasicKarel | $1$ | $4$ | $6.5$ |
> |  |  |  |  |
>
> -----
> **How much variation is there in the embeddings with the population? … can the performance prediction generalise to a different population?**
>
> Indeed, the agent population serves as an important component in our framework. Essentially, we gather task correlation statistics from an agent population that emerges during the multi-task policy learning process. For instance, please refer to our ablation experiment w.r.t. the population specification in Section 5.3, where agents in the population fail to learn to pick keys. This experiment shows how a poor-quality population affects both our proposed similarity criterion and the resulting task embedding space (see Fig. 4b).
>
> As previously mentioned, given a diverse population of agents, our proposed task similarity criterion captures performance correlations that remain valid for new agents. We demonstrate this in Appendix I by assessing the performance of the task embedding network in the performance prediction application scenario for agent populations trained differently.
>
> -----
>
> [He-Yueya et al., 2021] Quizzing Policy Using Reinforcement Learning for Inferring the Student Knowledge State. EDM 2021.
>
> [Ghosh et al., 2022] Adaptive Scaffolding in Block-based programming via Synthesizing New Tasks as Pop quizzes. AIED 2022.
>
> -----
>
> We hope that our responses address your concerns and are helpful in improving your rating. If you have any other comments or feedback, please let us know! We are looking forward to hearing back from you! Thank you again for the review.

---

> ### Author Response · Authors · 2023-11-21
>
> Dear Reviewer viG4,
>
> Since the Author/Reviewer Discussion period concludes on Nov. 22, we would be glad to address any remaining concerns or questions.
>
> Thanks,\
> Authors

---

> > ### Comment · Reviewer_viG4 · 2023-11-22
> >
> > Thanks very much for your answers to my questions. I have chosen to maintain my score. I remain concerned about the level of compute required for complex tasks and the applicability of the method given its requirement to train a population on a large range of tasks.

---

### Meta-Review · Area_Chair_LbFo · 2023-12-07

**Metareview:**

The authors introduce a task-embedding method based on a large number of trained agents. The main idea is to create a fixed size representation that capture similarity and difficulty levels of a range of different tasks.

As a positive, the reviewers found the paper easy to read and to understand.
Some reviewers also noted that the method beats relevant baselines.

However, there were a few key concerns:
First of all, multiple reviewers struggled to see the motivation for the method. This is particularly relevant since it requires large amounts of compute resources. Secondly, the reviewers would like to see comparison and connect of the method to prior work in representation learning methods for RL.
Lastly, one of the reviewers specifically asked for results on a more challenging benchmark (procgen, https://openai.com/research/procgen-benchmark) but the authors did not respond.

On balance, I think the paper would benefit from being resubmitted after addressing the key points raised.

**Justification For Why Not Higher Score:**

- computational requirements
- scalability to larger settings
- motivation for the method

**Justification For Why Not Lower Score:**

NA

---

### Decision · Program_Chairs · 2024-01-16

Reject